

# A filter design for T-S fuzzy systems based on moving horizon estimator with measurement noise

Hui Gao[1], Yixuan Wang[1] and Jing Hu[2]

[1] School of Electrical and Control Engineering, Shaanxi University of Science and Technology, Xi'an, China
[2] College of Information and Intelligent Transportation, Fujian Chuanzheng Communications College, Fuzhou, China

## ABSTRACT

In this article, a filter based on moving horizon estimator is proposed with Takagi-Sugeno (T-S) fuzzy controllers for a kind of unknown discrete-time system. The T-S fuzzy control algorithm is employed to handle the unknown system dynamics, thus ensuring the property of input-to-state stability (ISS) of the system, which guarantees the boundedness of all states. Besides, the proposed filter and controller can significantly improve the robustness of the system with external disturbance, even if the disturbance has non-Gaussian characteristics. Finally, the effectiveness of the presented algorithm is demonstrated by simulation examples under two kind of noise situations.

## INTRODUCTION

The Takagi-Sugeno (T-S) fuzzy model is a simple pattern to describe realistic systems, which has attracted vast interest of researchers in the systems and control field (*Su et al., 2012*; *Zeng et al., 2019*; *Yang, Xia & Liu, 2011*). Traditional fuzzy control systems are rule-based, which work well when there is no need to establish an reliable mathematical model for the system (*Dong, Wang & Gao, 2009*; *Nguang, Shi & Ding, 2007*). In contrast, T-S fuzzy patterns require mathematical expressions to represent the fuzzy results and reasoning under study. Filter designs for T-S fuzzy form are intended to estimate the system states by using the measured noise inputs so as to obtain the best estimation of unknown real signals or system states, and such designs have been considered useful in practical engineering aspects. The most common approach to resolve the problem of system state estimation, which has enjoyed wildly popularity, is the Kalman filter in the engineering field (*Anderson & Moore, 2012*; *Mendel, 1995*). However, the existing T-S fuzzy system is subject to various conditions when dealing with filtering problems, for example, the disturbance is Gaussian. It is essential to plan a filter that makes use of the data within a period of time instead of only the data at the previous moment to resolve the problem of the filtering process and improve the robustness of the T-S fuzzy system. This kind of filter can show a good effect in the T-S fuzzy method even without considering the form of external disturbance (*Ban et al., 2007*).

Corresponding author
Yixuan Wang,
200612052@sust.edu.cn

To investigate and synthesize nonlinear systems and hence depict complicated nonlinear relations, a T-S fuzzy control system is frequently utilized by establishing several simple linear relations (*Tseng, Chen & Uang, 2001*; *Xie et al., 2019*; *Chadli, Abdo & Ding, 2013*). It can also perform fuzzy reasoning and defuzzification on the outputs of several models. Many advances have been achieved in the study and control of T-S fuzzy systems in recent years. For example, to tackle the control problem for a type of nonlinear and unpredictable packet loss systems, a modified T-S fuzzy model was presented (*Dong, Wang & Gao, 2009*). In addition, the filtering problem of T-S fuzzy control scheme in discrete time system is studied, with the examples including $\ell_2$- $\ell_\infty$ filtering (*Su et al., 2012*), $H_\infty$ filtering (*Qiu, Feng & Yang, 2009*), and Kalman filtering (*Simon, 2003*; *Duncan & Horn, 2002*; *Bryson & Ho, 2018*). Kalman filtering is the most commonly used method to solve filtering and estimation problems in the T-S fuzzy systems, but it is suitable for linear systems (*Huang et al., 2017*). In other words, the applicability of Kalman filtering in T-S fuzzy system is limited by the need for a linear observation equation (*Kim & Bang, 2018*; *Goodwin et al., 1991*; *Sorenson, 1970*; *Box et al., 2015*). Also, the outliers of data sequence commonly affect the performance of Kalman filter (*Huber, 1992*).

These problems can be addressed by developing a moving horizon estimator (MHE) in the T-S fuzzy system. To our knowledge, there have been few studies on the use of MHE for solving the filtering problem for these nonlinear systems. Therefore, we expect this study will provide some important implications, both theoretical and practical, for this topic of research. As an online problem solving approach, MHE has been recognized to deal effectively with noise interference (*Yin & Gao, 2019*; *Rao, Rawlings & Lee, 2001*). Its basic idea is to use current measurements to update the optimization problem with the length of the time domain sliding window for processing data that remains unchanged (*Boulkroune, Darouach & Zasadzinski, 2010*; *Alessandri, Baglietto & Battistelli, 2003*). By applying the known state information for estimation, the rationality and accuracy of the estimation condition of the system will be considerably improved. In particular, if the MHE does not consider the time-domain constraints and the window length $N = 1$, it is the same as the Kalman filter (*Ling & Lim, 1999*). Over the past few decades, the MHE method has been widely investigated to support applications in several research areas. For example, it has been used to successfully address the estimation problem for the auto-regressive-moving-average with outliers contaminating the output (*Yin, Liu & Gao, 2018*; *Su et al., 2012*). The author uses the combination of MPC and T-S fuzzy system to design a predictive control method to solve the vehicle trajectory tracking problem, and uses the MHE to obtain the estimation of the vehicle state (*Alcala et al., 2020*). An MHE-based output feedback control algorithm is proposed and enables the overall system to converge to the origin (*Gharbi & Ebenbauer, 2021*). The authors introduce an MHE strategy to solve the estimation problem in a linear system with unknown input (*Zou et al., 2020*). Since MHE uses the states in a fixed-length time window to achive the filtering effect, this improves the robustness of the T-S fuzzy system and makes estimated value closer to ideal value.

The methods currently studied for the unknown discrete-time system usually use the T-S fuzzy control algorithm to deal with the unknown system dynamics. However, Kalman filter is often used in noise processing, but it has a very big limitation: it can only accurately

estimate linear process models and measurement models, and cannot achieve optimal estimation in nonlinear scenarios. The noise needs to have Gaussian characteristics. So we design a fuzzy controller filter based on the moving level estimator and guarantee the input-to-state stability (ISS) of the system, thus guaranteeing the boundedness of all states. Under the designed controller, the filter and controller can significantly improve the robustness of the external disturbance system even if the disturbance is non-Gaussian.

For a class of discrete systems with unknown disturbance, we present a filter based on MHE arithmetic and T-S fuzzy controller in this study. Firstly, for the studied system containing external interference, we established a T-S fuzzy model, systematically designed a filter based on MHE method, and obtained the relationship between estimated point and the points within the estimated window. Then, an optimal function with MHE constraints was proposed, so that the optimal solution satisfied the estimation relationship within a fixed-length time window. Finally, it was demonstrated that using MHE filters in the T-S fuzzy systems with bounded disturbance can guarantee input-to-state stability (ISS) characteristics.

The rest of the article is equipped as follows: Section 2 describes the prerequisite knowledge, including some definitions and basic properties of T-S fuzzy controllers. The main expressions and formulas as well as the method for finding the extreme value are introduced in Section 3. In Section 4, the ISS property of the T-S fuzzy system with MHE is proved. Section 5 indicates and discusses the simulation results of the pattern that we built. Finally, the conclusion is drawn in Section 6.

## PRELIMINARIES

An abundance of information on T-S fuzzy method and MHE has been provided in previous studies (*Dong, Wang & Gao, 2009*; *Tseng, Chen & Uang, 2001*; *Rao, Rawlings & Lee, 2001*; *Yin, Liu & Gao, 2018*; *Liu et al., 2016*). Obviously, approximating the nonlinear system to the form of a T-S fuzzy control system facilitates the subsequent processing. Therefore, in this section, the information required in the next section to derive the MHE with the measurement noise assumption is deduced, including the T-S fuzzy form representing the plants of the nonlinear systems and the MHE algorithm steps.

### Plant form

We think about a nonlinear device represented by way of a discrete-time T-S fuzzy model, as follows: Rule $i$: IF $\theta_{1,m}$ is $M_{i1}$ and ... and $\theta_{p,m}$ is $M_{ip}$, then

$$\begin{cases} x_{m+1} = A_i x_m + B_{2i} u_m + B_{1i} \omega_m \\ z_m = C_i x_m + D_{2i} u_m + \omega_m \\ x_m = \psi_m \end{cases} \tag{1}$$

where in the premise rules, $i = 1, 2, \ldots, r$, $\theta_m = [\theta_{1,m}, \theta_{2,m}, \ldots, \theta_{p,m}]$ is the premise variables vector, $M = [M_{i1}, M_{i2}, \ldots, M_{ip}]$ is the fuzzy set, $x_m \in \mathbb{R}^a$ is the state vector, $z_m \in \mathbb{R}^b$ is the measured output, $u_m \in \mathbb{R}^c$ is the input signal, $\omega_m \in \mathbb{R}^l$ represents the disturbance input vector, which is considered to be part of $l_2[0, \infty)$, and $r$ is the number of IF-THEN rules. $A_i, B_{1i}, B_{2i}, C_i, D_{2i}$ are known matrices with the appropriate dimensions.

The fuzzy basis functions are defined as follows:

$$h_i(\theta_m) = \frac{\prod_{j=1}^{p} M_{ij}(\theta_{j,m})}{\sum_{i=1}^{r} \prod_{j=1}^{p} M_{ij}(\theta_{j,m})} \tag{2}$$

where, for all $m$ values, we have $\prod_{j=1}^{p} M_{ij}(\theta_{j,m}) \geq 0$ ($i = 1, 2, \ldots, r$), and $\sum_{i=1}^{r} \prod_{j=1}^{p} M_{ij}(\theta_{j,m}) > 0$. Therefore, for all $m$ values the fuzzy basis functions satisfy the equations $h_i(\theta_m) \geqslant 0$ ($i = 1, 2, \ldots, r$) and $\sum_{i=1}^{r} h_i(\theta_m) = 1$.

Combine the fuzzy basis function with the proposed nonlinear system to get the following formula, which can be used for discrete systems under T-S fuzzy modeling:

$$\begin{cases} x_{m+1} &= \sum_{i=1}^{r} h_i(\theta_m)(A_i x_m + B_{2i} u_m + B_{1i} \omega_m) \\ z_m &= \sum_{i=1}^{r} h_i(\theta_m)(C_i x_k + D_{2i} u_m + \omega_m) \\ x_m &= \psi_m \end{cases} \tag{3}$$

For the convenience of calculation, we refer to experience to set the controller as a function related to the state feedback (*Dong, Wang & Gao, 2009*), that is, $u = kx$. Then Eq. (3) can be replaced by

$$\begin{cases} x_{m+1} &= \sum_{i=1}^{r} h_i(\theta_m)(\widehat{A}_i x_m + B_{1i} \omega_m) \\ z_m &= \sum_{i=1}^{r} h_i(\theta_m)(\widehat{C}_i x_m + \omega_m) \\ x_m &= \psi_m \end{cases} \tag{4}$$

where $\widehat{A}_i = A_i + kB_{2i}$, $\widehat{C}_i = C_i + kD_{2i}$. The MHE process for the T-S fuzzy system is still difficult to develop using this approach, so we further define

$$\overline{A}_m = \sum_{i=1}^{r} h_i(\theta_m)\widehat{A}_i, \quad \overline{B}_m = \sum_{i=1}^{r} h_i(\theta_m)B_{1i}$$

$$\overline{C}_m = \sum_{i=1}^{r} h_i(\theta_m)\widehat{C}_i, \quad \overline{D}_m = \sum_{i=1}^{r} h_i(\theta_m)$$

Here, we design the filters of a general structure by

$$\begin{cases} x_{m+1} &= \overline{A}_m x_m + \overline{B}_m \omega_m \\ z_m &= \overline{C}_m x_m + \overline{D}_m \omega_m \\ x_m &= \psi_m \end{cases} \tag{5}$$

The above formulas provide a great basis for our subsequent derivation.

## MHE for the T-S fuzzy model

Using the known information during this period of time such as $z_{m-L}, z_{m-L+1}, \ldots, z_m$ and $u_{m-L}, u_{m-L+1}, \ldots, u_m$ with the integer $L \geq 1$, we get the estimate through the MHE at time $m$. Using Eq. (5), we get the following formula between $x_{m+1}$ and $z_m$:

$$x_{m+1} = (\overline{A}_m - \overline{B}_m \overline{D}_m^{-1} \overline{C}_m)x_m + \overline{B}_m \overline{D}_m^{-1} z_m \tag{6}$$

For brevity, the following formula is used:

$$x_{m+1} = \Phi_m x_m + \Omega_m z_m \tag{7}$$

where $\Phi_m = \overline{A}_m - \overline{B}_m \overline{D}_m^{-1} \overline{C}_m$, $\Omega_m = \overline{B}_m \overline{D}_m^{-1}$. Using Eqs. (5) and (7), $L+1$ equations are iterated as shown below:

$$
\begin{aligned}
z_{m-L} &= \overline{C}_{m-L} x_{m-L} + \overline{D}_{m-L} \omega_{m-L} \\
z_{m-L+1} &= \overline{C}_{m-L+1} \Phi_{m-L} x_{m-L} + \overline{C}_{m-L+1} \Omega_{m-L} z_{m-L} + \overline{D}_{m-L+1} \omega_{m-L+1} \\
z_{m-L+2} &= \overline{C}_{m-L+2} \Phi_{m-L+1} \Phi_{m-L} x_{m-L} + \overline{C}_{m-L+2} \Phi_{m-L+1} \Omega_{m-L} z_{m-L} \\
&\quad + \overline{C}_{m-L+2} \Omega_{m-L+1} z_{m-L+1} + \overline{D}_{m-L+2} \omega_{m-L+2} \\
&\vdots \\
z_m &= \overline{C}_m \prod_{i=1}^{L} \Phi_{m-i} x_{m-L} + \overline{C}_m \sum_{j=1}^{L-1} \prod_{i=1}^{j} \Phi_{m-i} \Omega_{m-j-1} z_{m-j-1} + \overline{C}_m \Omega_{m-1} z_{m-1} + \overline{D}_m \omega_m
\end{aligned}
\tag{8}
$$

From Eq. (8) we know that the evaluate of measured output at the present time $m$ can be solved by the measured outputs at the time $m-1, m-2, m-3, \ldots, m-L$, the state of the system at the time $m-L$ and the measurement noise at the current time $m$.

***Remark 1:*** Here, we define $\overline{D}_m$ is an expression about fuzzy basis function in $\overline{D}_m = \sum_{i=1}^{r} h_i(\theta_m)$, without the coefficient matrix in the state space expression. Obviously, $\overline{D}_m$ here is an invertible matrix of dimension one.

***Remark 2:*** Kalman filter algorithm is based on accurate mathematical model and is sensitive to error. So the MHE in the T-S fuzzy system is proposed, which uses a fixed number of measurements for estimation. In this article, we derive a series of iterative formulas in order to obtain the relationship between $x_{m-L}$ and $z_{m-L}, z_{m-L+1}, \ldots, z_m$ within the fixed-length estimation window set by MHE.

## MAIN RESULTS

We introduce the simple expressions of explicit model by $Z_{m,L}$ and $W_{m,L}$, and propose an optimal function for the MHE. The output estimation of the T-S fuzzy system is taken as the target task, and the optimal value is obtained by a method in which the partial derivative is zero.

Using the second part of the recursive method, we define the following vectors:

$$
\begin{aligned}
Z_{m,L} &= [z_{m-L}^T, z_{m-L+1}^T, \ldots, z_{m-1}^T, z_m^T]^T \\
W_{m,L} &= [\omega_{m-L}^T, \omega_{m-L+1}^T, \ldots, \omega_{m-1}^T, \omega_m^T]^T
\end{aligned}
$$

and we assume that $\overline{Z}_{m,L} = T_L Z_{m,L}$, where

$$
T_L = \begin{bmatrix}
I & 0 & \cdots & 0 \\
-\overline{C}_{m-L+1} \Omega_{m-L} & I & \cdots & 0 \\
-\overline{C}_{m-L+2} \Phi_{m-L+1} \Omega_{m-L} & -\overline{C}_{m-L+2} \Omega_{m-L+1} & \cdots & 0 \\
\vdots & \vdots & \ddots & \vdots \\
-\overline{C}_m \left( \prod_{i=1}^{L-1} \Phi_{m-i} \right) \Omega_{m-L} & -\overline{C}_m \left( \prod_{i=1}^{L-2} \Phi_{m-i} \right) \Omega_{m-L+1} & \cdots & I
\end{bmatrix}
$$

and the equation for $\overline{Z}_{m,L}$ and $x_{m-L}$ can be written as

$$\overline{Z}_{m,L} = H_L x_{m-L} + E_L W_{m,L} \tag{9}$$

where

$$H_L = \begin{bmatrix} \overline{C}_{m-L} & 0 & \cdots & 0 \\ 0 & \overline{C}_{m-L+1}\Phi_{m-L} & \cdots & 0 \\ \vdots & \vdots & \ddots & \vdots \\ 0 & 0 & \cdots & \overline{C}_m\prod_{i=1}^{L}\Phi_{m-i} \end{bmatrix}$$

$$E_L = \begin{bmatrix} \overline{D}_{m-L} & 0 & \cdots & 0 \\ 0 & \overline{D}_{m-L+1} & \cdots & 0 \\ \vdots & \vdots & \ddots & \vdots \\ 0 & 0 & \cdots & \overline{D}_m \end{bmatrix}$$

The least squares criterion becomes the natural choice for deriving MHE when $\overline{x}_{m-L}$ is a priori prediction and $\sum_{m-L}$ is the corresponding covariance matrix. We define $\hat{x}_{m-L|m}$ as the estimation of $x_{m-L}$ at the time $m$. As a result, our goal at time $m$ is to determine the value of $\hat{x}_{m-L|m}$ which minimizes the following cost function $J$.

$$J = \|\overline{Z}_{m,L} - H_L x_{m-L}\|^2_{\Pi^{-1}_{m,L}} + \|\hat{x}_{m-L|m} - \overline{x}_{m-L}\|^2_{\Sigma^{-1}_{m-L}} \tag{10}$$

where

$$\Pi^{-1}_{m,L} = \begin{bmatrix} R_{m-L} & 0 & \cdots & 0 \\ 0 & R_{m-L+1} & \cdots & 0 \\ \vdots & \vdots & \ddots & \vdots \\ 0 & 0 & \cdots & R_m \end{bmatrix}$$

From formula Eq. (8), once the value of $\hat{x}_{m-L|m}$ is obtained, we can get the value of $\hat{x}_{m-L+1|m}, \hat{x}_{m-L+2|m}, \ldots, \hat{x}_{m|m}$ by

$$\hat{x}_{i+1|m} = \Phi_m \hat{x}_{i|m} + \Omega_m z_i \tag{11}$$

with $i = m-L, m-L+1, m-L+2, \ldots, m-1$, so that the estimation of output $\hat{z}_{m|m}$ can be solved by

$$\hat{z}_{m|m} = \overline{C}_m \hat{x}_{m|m} \tag{12}$$

A variety of methods can be used to obtain the prior prediction $\overline{x}_{m-L}$ of the cost function. In this article, the most common method is used, which is expressed as follows *Camacho & Alba (2013)*:

$$\overline{x}_{m-L} = \Phi_{m-L-1}\hat{x}_{m-L-1|m-L-1} + \Omega_{m-L-1} z_{m-L-1} \tag{13}$$

Corresponding to Eq. (13), the correlation covariance $\Sigma_{m-L}$ satisfies the following:

$$\Sigma_{m-L} = \Phi_{m-L-1} P_{m-L-1|m-L-1} \Phi^T_{m-L-1} \tag{14}$$

For Eq. (10), the smaller the cost function $J$ is, the closer the estimated value is to the true value.

## STABILITY ANALYSIS

With the bounded external signal input, if the state response is within the bounded range, the system satisfies ISS. In other words, if any external input and initial conditions are bounded, the state bounded. The system will always have the ability to return to the equilibrium point when the external input is 0.

### Input-to-state stability (ISS)

Non-linear systems with external disturbances are considered as follows:

$$x_{m+1} = \overline{A}_m x_m + \overline{B}_m \omega_m. \tag{15}$$

Here, we provide two ISS definitions.

**Lemma 1** *Alessandri, Baglietto & Battistelli (2008)* The system in Eq. (15) is input-to-state stable (ISS) if there exist the function $\beta \in KL$ and the function $\gamma \in K_\infty$ such that for each external input $\omega(m)$ and each initial condition $x_0 = \overline{x}_{m-L}$, solutions exist and satisfy

$$\|x_{m,x_0,\omega_m}\| \leq \beta(\|x_0,L\|) \leq \gamma(\omega_m) \tag{16}$$

where $\|x_{m,x_0,\omega_m}\|$ is the solution to the system in Eq. (15) at time $m$.

**Lemma 2** *Kim et al. (2006)* The system in Eq. (15) is input-to-state stable (ISS) if and only there exists the continuous ISS-Lyapunov function $V: R^n \to R \geq 0$ such that for the functions $\lambda_1, \lambda_2, \lambda_3, \sigma \in K_\infty$, the Lyapunov function $V$ satisfies

$$\lambda_1 \|x_m\| \leq V(x_m) \leq \lambda_2 \|x_m\| \tag{17}$$

and

$$V(x_{m+1}) - V(x_m) \leq -\lambda_3 \|x_m\| + \sigma \|\omega_m\| \tag{18}$$

or

$$V(x_{m+1}) - V(x_m) \leq -\lambda_3 \|x_{m+1}\| + \sigma \|\omega_m\| \tag{19}$$

### ISS of the proposed MHE

Before proving ISS of the system in Eq. (15) under the MHE, we need to calculate the estimation of $x_{m-L}$ considering the cost function $J$ at time $m$ having the smallest value, such that the cost function $J$ satisfies

$$\frac{\partial J}{\partial \hat{x}_{m-L|m}} = 0 \tag{20}$$

By calculation, we obtain the equation for $\hat{x}_{m-L|m|}$ as follows:

$$2\Sigma_{m-L}^{-1}(\hat{x}_{m-L|m} - \overline{x}_{m-L}) = 0 \tag{21}$$

Using Eq. (21), the solution can be obtained by

$$\hat{x}_{m-L|m} = \overline{x}_{m-L}. \tag{22}$$

This subsection introduces the stability characteristics of the estimation error of the proposed unconstrained estimator. Using Eq. (22), the estimated error $e_{m-L}$ is given as follows:

$$
\begin{aligned}
e_{m-L} = & \; x_{m-L} - \hat{x}_{m-L|m} \\
= & \; x_{m-L} - \bar{x}_{m-L} \\
= & \; \Phi_{m-L-1} x_{m-L-1} + \Omega_{m-L-1} z_{m-L-1} - \Phi_{m-L-1} \hat{x}_{m-L-1|m-L-1} - \Omega_{m-L-1} z_{m-L-1}
\end{aligned}
\tag{23}
$$

Then, we get the estimated error dynamics:

$$
e_{m-L} = \Phi_{m-L-1} e_{m-L-1}.
\tag{24}
$$

The pair $(\overline{C}_m, \overline{A}_m)$ is completely observable in $L$ step.

**Theorem**: Consider a pair $\{\bar{x}_{m-L} \text{ and } Z_{m,L}\}$ and suppose that Assumption 1 holds. If there exists a scaler $\mu$ and symmetric matrices $P_1 > 0$, $P_2 > 0$ satisfy

$$
\|\Phi_{m-L-1}\| < 1
\tag{25}
$$

$$
P_2 - P_1 \leq -Q_1
\tag{26}
$$

$$
P_2 - P_1 \geq -Q_2
\tag{27}
$$

for some $Q_1 > 0, Q_2 > 0$, then the estimation error dynamics $e_{m-L}$ are ISS.

**Proof:** If $\|\Phi_{m-L-1}\| < 1$, then $\rho(\Phi_{m-L-1}) < 1$ is obtained, that means that there is always a matrix $P_1$ that satisfies

$$
\Phi_{m-L-1}^T P_1 \Phi_{m-L-1} - P_1 \leq -Q_1
\tag{28}
$$

for any $Q_1 = Q_1^T > 0$. Simple algebraic manipulations show that

$$
\|\Phi_{m-L-1} e_{m-L-1}\|_{P_1}^2 - \|e_{m-L-1}\|_{P_1}^2 \leq -\|e_{m-L-1}\|_{Q_1}^2
\tag{29}
$$

Using Eq. (24), the following equality can be obtained:

$$
\|\Phi_{m-L-1} e_{m-L-1}\|_{P_1}^2 - \|e_{m-L-1}\|_{P_1}^2 = \|e_{m-L}\|_{P_1}^2
\tag{30}
$$

Combining Eqs. (29) and (30) yields

$$
\|e_{m-L}\|_{P_1}^2 \leq \|e_{m-L-1}\|_{Q_1 - P_1}^2
\tag{31}
$$

Consider the Lyapunov candidate $V$: $V(e_{m-L}) = \|e_{m-L}\|_{P_2}^2$, then

$$
\begin{aligned}
& V(e_{m-L}) - V(e_{m-L-1}) \\
= & \|e_{m-L}\|_{P_2}^2 - \|e_{m-L-1}\|_{P_1}^2 \\
\leq & \|e_{m-L}\|_{P_2}^2 - \|e_{m-L}\|_{P_1}^2 \\
\leq & \|e_{m-L}\|_{P_2 - P_1}^2 \\
\leq & -\|e_{m-L}\|_{Q_2}^2 \\
\leq & -\delta\|e_{m-L}\|
\end{aligned}
\tag{32}
$$

where $\delta = \frac{1}{2}\lambda_{min}(Q_2) r^2$. As a result, Theorem 1 is derived. The ISS analysis result is presented in Eq. (15).

## SIMULATION AND EXPERIMENTS

To validate the aforementioned statements, the control problem for some examples of the proposed MHE is considered.

Considering the T-S fuzzy system in Eq. (4), we know

$$
\begin{cases}
x_{m+1} &= \sum_{i=1}^{r} h_i(\theta_m)(\widehat{A}_i x_m + B_{1i} \omega_m) \\
z_m &= \sum_{i=1}^{r} h_i(\theta_m)(\widehat{C}_i x_m + \omega_m)
\end{cases}
\tag{33}
$$

Assume that $\theta_m \in [-M, M]$ and $M > 0$. The nonlinear term $\theta_m^2$ can be accurately expressed as *Su et al. (2012)*

$$
\theta_m^2 = h_1(\theta_m)(-M)\theta_m + h_2(\theta_m)M\theta_m
\tag{34}
$$

where $h_1(\theta_m), h_2(\theta_m) \in [0, 1]$ and $h_1(\theta_m) + h_2(\theta_m) = 1$. Through the above equations, the membership functions $h_1(\theta_m)$ and $h_2(\theta_m)$ are solved as

$$
h_1(\theta_m) = \frac{1}{2} - \frac{\theta_m}{2M}, h_2(\theta_m) = \frac{1}{2} + \frac{\theta_m}{2M}
\tag{35}
$$

The following conclusion can be obtained from the above expressions that $h_1(\theta_m) = 1$ and $h_2(\theta_m) = 0$ when $\theta_m$ is $-M$ and that $h_1(\theta_m) = 0$ and $h_2(\theta_m) = 1$ when $\theta_m$ is $M$. Then, to approximate the nonlinear system, the T-S fuzzy model suggested below can be used:

*plant form:*

*Rule 1*: IF $\theta_k = -M$, THEN

$$
\begin{cases}
x_{m+1} &= \widehat{A}_1 x_m + B_{11} \omega_m \\
z_m &= \widehat{C}_1 x_m + \omega_m
\end{cases}
$$

*Rule 2*: IF $\theta_m = M$, THEN

$$
\begin{cases}
x_{m+1} &= \widehat{A}_2 x_m + B_{12} \omega_m \\
z_m &= \widehat{C}_2 x_m + \omega_m
\end{cases}
$$

and the following are the system matrices:

$$
\widehat{A}_1 = \begin{bmatrix} AM & 0.1 \\ A & 0 \end{bmatrix}, B_{11} = \begin{bmatrix} 1 \\ 0 \end{bmatrix}, \widehat{C}_1 = \begin{bmatrix} A & 0 \end{bmatrix}
$$

$$
\widehat{A}_2 = \begin{bmatrix} -AM & 0.1 \\ A & 0 \end{bmatrix}, B_{12} = \begin{bmatrix} 1 \\ 0 \end{bmatrix}, \widehat{C}_1 = \begin{bmatrix} A & 0 \end{bmatrix}
$$

In the example, $x_m = [x_{1,m}^T x_{2,m}^T]^T$, $A = 0.6$, $M = 0.2$, so that

$$
\widehat{A}_1 = \begin{bmatrix} 0.12 & 0.1 \\ 0.6 & 0 \end{bmatrix}, B_{11} = \begin{bmatrix} 1 \\ 0 \end{bmatrix}, \widehat{C}_1 = \begin{bmatrix} 0.6 & 0 \end{bmatrix}
$$

$$\widehat{A}_2 = \begin{bmatrix} -0.12 & 0.1 \\ 0.6 & 0 \end{bmatrix}, B_{12} = \begin{bmatrix} 1 \\ 0 \end{bmatrix}, \widehat{C}_1 = \begin{bmatrix} 0.6 & 0 \end{bmatrix}$$

The proposed method uses simulation and experimental data to test performance. We present an algorithm that summarizes the steps involved in the MHE proposed in the T-S fuzzy system. For some intermediate steps, we need to repeat some calculation formulas cyclically.

After research, our algorithm process is following:

## Algorithm

- Give the initial values $x_0$ and set $L = 5$.
- Establish T-S fuzzy control system model Eq. (22).
- Solve $x_m$ and $z_m$ in the form of the system.
- Solve $\Phi_m$ and $\Omega_m$ by formula Eq. (7).
- Obtain the prior prediction $\overline{x}_{m-L}$ by formula Eq. (13).
- Calculate the estimation $\hat{x}_{m-L|m|}$ so that $\hat{x}_{m|m|}$ and $\hat{z}_{m|m|}$ using the MHE.
- Set $m = m + 1$ and go back to step 5.
- Get the estimated value of all state data and end the algorithm.

In the T-S fuzzy control system, two different noise conditions are given to verify the effect of the proposed MHE. The first case is that the noise function is given as the noise gradually decreases over time, and the other case is that the noise is Gaussian noise.

Case 1 (Gaussian noise):

Let the initial condition be zero, that is, $x_0 = 0$ ($\hat{x}_0 = 0$), and suppose the disturbance input $\omega_k$ is $N(0, 1)$. Under the above-mentioned setting conditions, in order to better illustrate the universality that MHE can achieve the goal, we randomly select Gaussian noise and obtain the estimation result using MHE of the system.

The estimation result of the T-S fuzzy system with Gaussian noise is shown in Fig. 1. Obviously, under the influence of Gaussian noise, the output of the system changes more widely, and the output after adding MHE is more gradual. It shows that when the measured noise satisfies the normal distribution, the performance of estimation is remarkable, and the estimated value curve fluctuates within a smaller range than the true value curve.

Case 2 (non-Gaussian noise):

Let the initial condition $x_0 = 0$ ($\hat{x}_0 = 0$), and assume the disturbance input $\omega_m$ is

$$\omega_m = \frac{3 * sin(0.85m)}{(0.55m)^2 + 1} \tag{36}$$

The simulation results are shown in Figs. 2 and 3. Figure 2 is the noise, obviously, the external interference is bounded and non-Gaussian. Figure 3 shows the simulation run for the T-S fuzzy system with the MHE filter. The proposed MHE can effectively counteract the influence of the sine-form noise in the T-S fuzzy system. In this case, the noise decays with time, and the estimation performance of the MHE is most pronounced during the initial period. A clear improvement of the smoothness can be observed for the T-S fuzzy system, which is the result of the MHE filter reducing noise.

Case 3 (non-Gaussian noise):

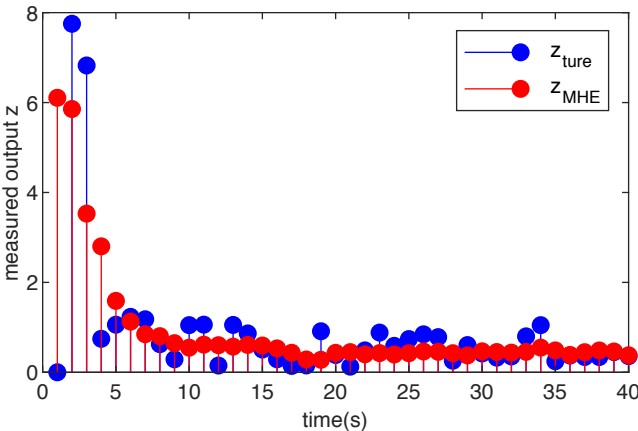

**Figure 1** The true measured output $z(m)$ and its estimations $\hat{z}(m)$ based on the MHE with Gaussian noise.

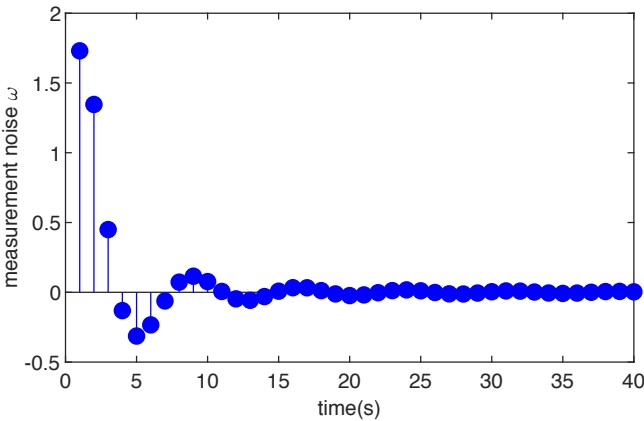

**Figure 2** The noise of T-S fuzzy system in case 2.

To make our proposed MHE estimation scheme more convincing in systems with unknown dynamics, we add a case when the noise is uniformly distributed.

The added noise in this case is shown in Fig. 4 and the effect of the designed MHE filter is shown in Fig. 5. It can be seen from the figure that adding the MHE filter to the T-S fuzzy control model with uniformly distributed noise can make the output smoother. To increase the convincing power, a uniformly distributed noise is added to the designed multi-threaded control system, and MHE filtering is used. It can be seen from the simulation figures that the proposed estimator can work well in systems with unknown factors.

Through the above two kinds of different noise simulations, we find that it is feasible to use MHE to solve the discrete-time filtering problem. The filter based on the MHE method we designed shows a good effect in the T-S fuzzy system with external disturbance, even if the disturbance is non-Gaussian.

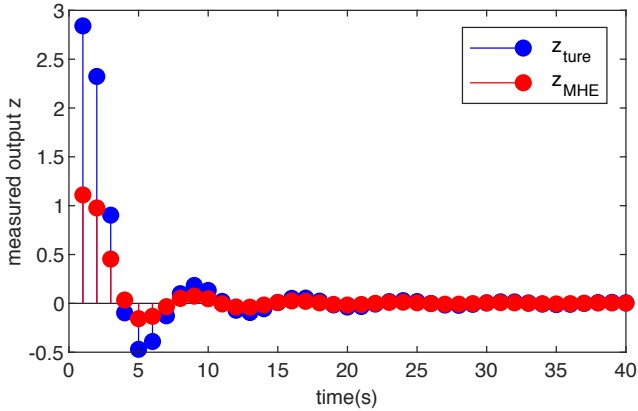

**Figure 3** The true measured output $z(m)$ and its estimations $\hat{z}(m)$ based on the MHE with function noise.

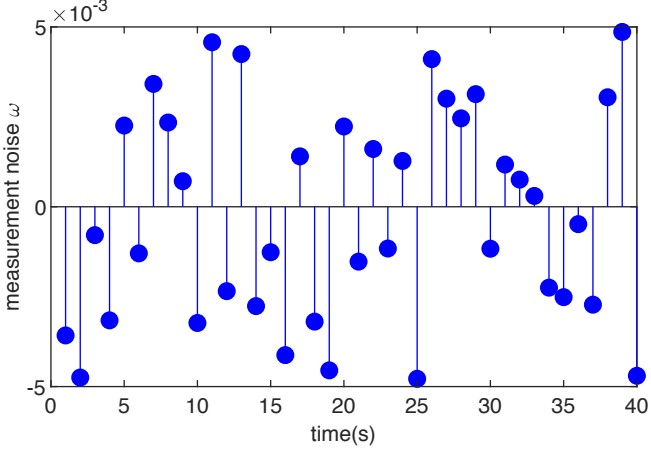

**Figure 4** The noise of T-S fuzzy system in case 3.

## CONCLUSIONS

This article presents a design to solve the filtering problem for the performance of MHE in discrete-time T-S fuzzy systems. An MHE different from the traditional Kalman filter was proposed. At first, a presentation mode of the discrete time system was employed to convert the authentic machine into T-S fuzzy system. Based on the T-S fuzzy model, the proposed MHE was used to obtain a more precise estimate for the filtering error system. Then, the analytical solution for the proposed MHE as well as the result when the cost function has the smallest value was obtained. Next, the ISS property of the proposed MHE was examined. Finally, the proposed method was demonstrated to be effective by simulation examples.

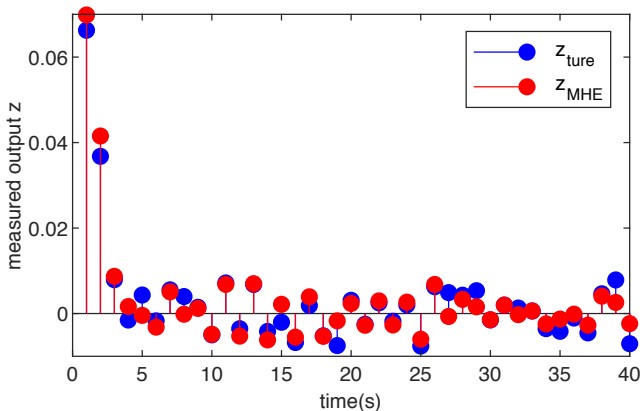

**Figure 5** The true measured output $z(m)$ and its estimations $\hat{z}(m)$ based on the MHE with uniform noise.

### Funding

This work was supported by the Natural Science Foundation of Shaanxi under Grant 2022JQ-651. The funders had no role in study design, data collection and analysis, decision to publish, or preparation of the manuscript.

### Grant Disclosures

The following grant information was disclosed by the authors:
Natural Science Foundation of Shaanxi: 2022JQ-651.

### Competing Interests

The authors declare there are no competing interests.

### Author Contributions

- Hui Gao conceived and designed the experiments, performed the experiments, analyzed the data, performed the computation work, prepared figures and/or tables, authored or reviewed drafts of the article, and approved the final draft.
- Yixuan Wang conceived and designed the experiments, performed the experiments, analyzed the data, performed the computation work, prepared figures and/or tables, authored or reviewed drafts of the article, and approved the final draft.
- Jing Hu performed the experiments, analyzed the data, performed the computation work, prepared figures and/or tables, authored or reviewed drafts of the article, and approved the final draft.

### Data Availability

The results of the measurement error and T-S Fuzzy system with MHE filtering are available in the Supplemental Files.

## Supplemental Information

Supplemental information for this article can be found online at http://dx.doi.org/10.7717/peerj-cs.1208#supplemental-information.

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
