# Peer review of "A filter design for T-S fuzzy systems based on moving horizon estimator with measurement noise"

_PeerJ Computer Science, doi:10.7717/peerj-cs.1208_

## Round 0.1 · original submission · Major Revisions

Please make sure Reviewer 2's comments, especially regarding remark 2 are appropriately addressed.

·

Basic reporting

Authors are suggested to include more recent and relevant research, especially from the last five years to increase the knowledge base of the manuscript.

Experimental design

More results should be added to highlight the contributions of the manuscript.

Validity of the findings

The results should be compared with the other literature works to ensure the validity of the work.

Additional comments

This is an interesting work on the design of the Moving Horizon Estimator filter with the TS fuzzy controllers. The input-to-state stability is well derived for the proposed MHE. simulation runs are presented for non-Gaussian and Gaussian noises. After reading carefully, it is observed that the manuscript could be accepted with few revisions such as

1. The literature should be strengthened with recent works.
2. The contributions of the work should be clearly highlighted, preferably, point-wise before the organization of the manuscript in the 'Introduction' Section.
3. Authors should present the proper reasoning behind the selection of the Gaussian noises. There are other more critical noises such as 'speckle noise', 'Poison noise', etc.
4. The simulation results should be compared to the existing works in the literature, quantitatively or qualitatively.
5. Why the equations in Simulation and Experiments are not marked? The section 'Simulation and Experiments' should be renamed. I could not find any experiment results.

Reviewer 2 ·

Basic reporting

In the paper the authors proposed a filter using T-S fuzzy controllers for discrete systems. At first it is not clear what class of discrete systems they are referring to. The Abstract is somehow vague and needs improvement.


First page - please correct “ffiilter”

Section Introduction - I believe “vast” is more appropriated than “giant”

Section Introduction - I wouldn’t use “hassle” but “problem”. The colloquial use of the language should be avoided. There are several instance in the manuscript where the author made use of colloquial words where there is proper technical words to describe the same thing.

Remark 2 - The comment on MHE does not seem to solve the problem pointed out in the first sentence of the remark. So does MHE come as the solution of the problem of accurate mathematical model and sensitivity to error?

Zm,l and Wm,l are mentioned before establishing their definition.

Equation 13 - Please give References to the method and explicitly define the most common method. The reader should be able to replicate all derivations.

Just before Equation 21, it is not clear what the authors meant by “By calculation”and how they did it.

Section “Simulation and Experiments”- From my point of view, this section needs to be throughly revised. Where did the nonlinear term came from? Why L=5? What are the values of the initial conditions? And so on.

The major problem in this section is that the figures seem to contract what the authors said in the text. The authors need to explain what exactly is remarkable about the results presented in the paper. The figures showed exactly the contrary and are not convincing (std = 0.0001 - almost no noise contamination - please add a convincing noise threshold).

Experimental design

See above

Validity of the findings

See above

Additional comments

In summary, I believe that there is a potential contribution work publishing but the manuscript should undergo a second round of reviews.

---

## Round 0.2 · accepted · Accept

The authors have addressed all of the reviewers' comments.

·

Basic reporting

No comment

Experimental design

No comment

Validity of the findings

No comment

Additional comments

Authors have addressed most of the concerns raised by the reviewer. The manuscript can be accepted for publication as it has significantly improved after revision.

---

## Author Rebuttal · Round 0.2

# Response to the Comments by the Editor, Associate Editor and Reviewers

**Title:** A filter design for T-S fuzzy systems based on moving horizon estimator with measurement noise.

**Authors:** Hui Gao[1], Yixuan Wang[1,*], Jing Hu[2]

1, College of Electrical and Control Engineering,
Shaanxi University of Science and Technology, China.
2, College of Information and Intelligent Transportation,
Fujian Chuanzheng Communications College, China.

**Manuscript ID:** CS-2022:06:75039:0:1:

# 1, Reply to Editor, Associate Editor and Reviewers:

***Reply:*** We are thankful to all of you for handling this paper and providing insightful and constructive comments. Based on your recommendation and suggestions, we have made the revision carefully and explained the detail point-by-point in the following replies. For your reference, some modified parts have been highlighted in blue in revised manuscript.

# 2, Reply to Reviewer 1:

**1.Basic reporting.** Authors are suggested to include more recent and relevant research, especially from the last five years to increase the knowledge base of the manuscript.

***Reply:*** *Thank you for your professional suggestion. We have added some recent research on MHE in the third paragraph of the Introduction section, to increase the knowledge base of the manuscript.*

*The author uses the combination of MPC and T-S fuzzy system to design a predictive control method to solve the vehicle trajectory tracking problem, and uses the MHE to obtain the estimation of the vehicle state Alcala et al. (2020). An MHE-based output feedback control algorithm is proposed and enables the overall system to converge to the origin Gharbi and Ebenbauer (2021). The authors introduce an MHE strategy to solve*

the estimation problem in a linear system with unknown input Zou et al. (2020).

[1] Alcala, E., Sename, O., Puig, V., and Quevedo, J. (2020). Ts-mpc for autonomous vehicle using a learning approach. IFAC-PapersOnLine, 53(2):15110–15115.

[2] Gharbi, M. and Ebenbauer, C. (2021). Anytime mhe-based output feedback mpc. IFAC-PapersOnLine, 54(6):264–271.

[3] Zou, L., Wang, Z., Hu, J., and Zhou, D. (2020). Moving horizon estimation with unknown inputs under dynamic quantization effects. IEEE Transactions on Automatic Control, 65(12):5368–5375.

**2.Experimental design.** More results should be added to highlight the contributions of the manuscript.

**Reply:** *Thank you for your professional suggestion. The contribution has been updated in the revised version. The methods currently studied for the unknown discrete-time system usually use the T-S fuzzy control algorithm to deal with the unknown system dynamics. However, Kalman filter is often used in noise processing, but it has a very big limitation: it can only accurately estimate linear process models and measurement models, and cannot achieve optimal estimation in nonlinear scenarios. And the noise needs to have Gaussian characteristics. So we design a fuzzy controller filter based on the moving level estimator and guarantee the input-to-state stability (ISS) of the system, thus guaranteeing the boundedness of all states. Under the designed controller, the filter and controller can significantly improve the robustness of the external disturbance system even if the disturbance is non-Gaussian.*

**3.Validity of the findings.** The results should be compared with the other literature works to ensure the validity of the work.

**Reply:** *Thank you for your excellent suggestion. In the existing research, if the system with unknown dynamics contains Gaussian noise, the Kalman filter can be used to process the output value. We compared the effect of MHE filter and Kalman filter and found that MHE can get a smoother estimate and the effect is better.*

[Figure]

Figure 1: Comparison of effects of different filters in T-S fuzzy fuzzy control system with Gaussian noise

**4.Additional comments.** This is an interesting work on the design of the Moving Horizon Estimator filter with the T-S fuzzy controllers. The input-to-state stability is well derived for the proposed MHE. simulation runs are presented for non-Gaussian and Gaussian noises. After reading carefully, it is observed that the manuscript could be accepted with few revisions such as

1. The literature should be strengthened with recent works.

2. The contributions of the work should be clearly highlighted, preferably, point-wise before the organization of the manuscript in the 'Introduction' Section.

3. Authors should present the proper reasoning behind the selection of the Gaussian noises. There are other more critical noises such as 'speckle noise', 'Poison noise', etc.

4. The simulation results should be compared to the existing works in the literature, quantitatively or qualitatively.

5. Why the equations in Simulation and Experiments are not marked? The section 'Simulation and Experiments' should be renamed. I could not find any experiment results.

*Reply: Thank you for your careful reading. Regarding the above issues, we have made serious changes in the revised version:*

*1. We have added nearly 5 years of research on moving horizon estimators in the revised edition to facilitate readers' better understanding. The author uses the combination*

*of MPC and T-S fuzzy system to design a predictive control method to solve the vehicle trajectory tracking problem, and uses the MHE to obtain the estimation of the vehicle state Alcala et al. (2020). An MHE-based output feedback control algorithm is proposed and enables the overall system to converge to the origin Gharbi and Ebenbauer (2021). The authors introduce an MHE strategy to solve the estimation problem in a linear system with unknown input Zou et al. (2020).*

*2. We refine the research motivation in the introduction chapter of the revised edition. Our research motivation: The contribution has been updated in the revised version. The methods currently studied for the unknown discrete-time system usually use the T-S fuzzy control algorithm to deal with the unknown system dynamics. However, Kalman filter is often used in noise processing, but it has a very big limitation: it can only accurately estimate linear process models and measurement models, and cannot achieve optimal estimation in nonlinear scenarios. And the noise needs to have Gaussian characteristics. So we design a fuzzy controller filter based on the moving level estimator and guarantee the input-to-state stability (ISS) of the system, thus guaranteeing the boundedness of all states. Under the designed controller, the filter and controller can significantly improve the robustness of the external disturbance system even if the disturbance is non-Gaussian.*

*3. In our study of noise processed by the moving horizon estimator, we compared Gaussian noise and non-Gaussian noise. The purpose is that the designed estimator can achieve better estimation results than the Kalman filter in the case of establishing a T-S fuzzy model in a system with unknown system dynamics. The poisonous noise and speckle noise mentioned are often present in the image processing process, not in the process of the control system we mentioned. For the studied systems containing non-Gaussian noise, we have enriched the simulation section for better understanding by the reader.*

*4. Regarding the simulation experiment part, we have improved the previous work in the revised version:*

*case 1 (Gaussian noise): Let the initial condition be zero, that is, $x_0 = 0$ ($\hat{x}_0 = 0$), and suppose the disturbance input $\omega_m$ is $N(0, 1)$. Under the above-mentioned setting conditions, in order to better illustrate the universality that MHE can achieve the goal, we randomly select Gaussian noise and obtain the estimation result using MHE of the system.*

[Figure]

Figure 2: The true measured output $z(m)$ and its estimations $\hat{z}(m)$ based on the MHE with Gaussian noise

*The estimation result of the T-S fuzzy system with Gaussian noise is shown in Figs. 2.*

*case 2 (non-Gaussian noise): Let the initial condition $x_0 = 0$ ($\hat{x}_0 = 0$), and assume the disturbance input $\omega_m$ is*

$$\omega_m = \frac{3 * sin(0.85m)}{(0.55m)^2 + 1}$$

[Figure]

Figure 3: The noise of T-S fuzzy system in case 2

[Figure]

Figure 4: The true measured output $z(m)$ and its estimations $\hat{z}(m)$ based on the MHE with function noise

*The simulation results are shown in Figs. 3 and 4. Fig. 3 is the noise, obviously, the external interference is bounded and non-Gaussian. And Fig. 4 shows the simulation run for the T-S fuzzy system with the MHE filter.*

*case 3 (non-Gaussian noise): To make our proposed MHE estimation scheme more convincing in systems with unknown dynamics, we add a case when the noise is uniformly distributed.*

[Figure]

Figure 5: The noise of T-S fuzzy system in case 3

[Figure]

Figure 6: The true measured output $z(m)$ and its estimations $\hat{z}(m)$ based on the MHE with uniform noise

The added noise in this case is shown in Fig. 5 and the effect of the designed MHE filter is shown in Fig. 6. It can be seen from the figure that adding the MHE filter to the T-S fuzzy control model with uniformly distributed noise can make the output smoother.

5. In the revised version, we have labelled the formulas in the simulation for the convenience of the reader and sorted the formulas in the simulation part. And make a clearer description of the conclusions drawn from the simulation part:

case 1 : Obviously, under the influence of Gaussian noise, the output of the system changes more widely, and the output after adding MHE is more gradual. It shows that when the measured noise satisfies the normal distribution, the performance of estimation is remarkable, and the estimated value curve fluctuates within a smaller range than the true value curve.

case 2 : The proposed MHE can effectively counteract the influence of the sine-form noise in the T-S fuzzy system. In this case, the noise decays with time, and the estimation performance of the MHE is most pronounced during the initial period. A clear improvement of the smoothness can be observed for the T-S fuzzy system, which is the result of the MHE filter reducing noise.

case 3 : To increase the convincing power, a uniformly distributed noise is added to the designed multi-threaded control system, and MHE filtering is used. It can be seen from the simulation figures that the proposed estimator can work well in systems with unknown

*factors.*

# 3, Reply to Reviewer 2:

**1.Basic reporting.** In the paper the authors proposed a filter using T-S fuzzy controllers for discrete systems. At first it is not clear what class of discrete systems they are referring to. The Abstract is somehow vague and needs improvement.

*Reply: Thank you for your excellent suggestion. In this paper, we consider the discrete system represented by a siscrete-time T-S fuzzy model. In fact, many kind of nonlinear discrete system can be approximated by the T-S fuzzy model, which shows that the proposed method in this paper can be applied to many different system. The Abstract and the Introduction are revised to improve the readability and highlight the contribution.*

**2.First page.** please correct "ffiilter".

*Reply: Thank you for your careful reading. We have carefully fixed spelling errors in the revised version: In this paper, a filter based on Moving Horizon Estimator is proposed with Takagi-Sugeno (T-S) fuzzy controllers for a kind of unknown discrete-time system.*

**3.Section Introduction.** I believe "vast" is more appropriated than "giant". I wouldn't use "hassle" but "problem". The colloquial use of the language should be avoided. There are several instance in the manuscript where the author made use of colloquial words where there is proper technical words to describe the same thing.

*Reply: You make a good point. We have corrected colloquial words in the revised version and replaced them with appropriate words:*

*Takagi-Sugeno (T-S) fuzzy model is a simple pattern to describe realistic systems, which has attracted vast interest of researchers in the systems and control field Su et al. (2012); Zeng et al. (2019); Yang et al. (2011).*

*And the most normally used approach to resolve the problem of system state estimation, which has enjoyed wildly popularity, is the Kalman filter in the engineering field Anderson and Moore (2012); Mendel (1995).*

*For example, to tackle the control problem for a type of nonlinear and unpredictable packet loss systems, a modified T-S fuzzy model was presented Dong et al. (2009).*

**4.Remark 2.** The comment on MHE does not seem to solve the problem pointed out in the first sentence of the remark. So does MHE come as the solution of the problem of accurate mathematical model and sensitivity to error?

*Reply: Thank you for your careful reading. In the presence of modeling errors, the Kalman filter may give highly inaccurate estimation results and eventually diverge as the errors are accumulated. By contrast, the MHE only makes use of the most recent N measurements and avoids accumulation of modeling errors. Hence, better estimation results can be expected if MHE is used.*

**5.** Zm,l and Wm,l are mentioned before establishing their definition.

*Reply: Thank you for your careful reading. We introduce the simple expressions of explicit model by $Z_{m,L}$ and $W_{m,L}$, and propose an optimal function for the MHE. $Z_{m,L}$ is the matrix associated with the outputs $z_{m-L}, ..., z_m$ and $W_{m,L}$ is the matrix associated with the noise $\omega_{m-L}, ..., \omega_m$. Using the second part of the recursive method, we define the following vectors:*

$$Z_{m,L} = [z_{m-L}^T, z_{m-L+1}^T, \cdots, z_{m-1}^T, z_m^T]^T$$
$$W_{m,L} = [\omega_{m-L}^T, \omega_{m-L+1}^T, \cdots, \omega_{m-1}^T, \omega_m^T]^T$$

**6.Equation 13.** Please give References to the method and explicitly define the most common method. The reader should be able to replicate all derivations.

*Reply: Thank you for your helpful suggestion. For ease of understanding, we have added relevant references in the revised edition.*

*[4] Chadli, M., Abdo, A., and Ding, S. X. (2013). $h-/h\infty$ fault detection filter design for discrete-time takagi–sugeno fuzzy system. Automatica, 49(7):1996–2005.*

**7.** Just before Equation 21, it is not clear what the authors meant by "By calculation" and how they did it.

*Reply: Thank you for your professional suggestion. The expression of the cost function is $J = \|\overline{Z}_{m,L} - H_L x_{m-L}\|^2_{\Pi_{m,L}^{-1}} + \|\hat{x}_{m-L|m} - \overline{x}_{m-L}\|^2_{\Sigma_{m-L}^{-1}}$, the necessary condition for its value to reach the minimum value is $\dfrac{\partial J}{\partial \hat{x}_{m-L|m}} = 0$. So we can get $2\Sigma_{m-L}^{-1}(\hat{x}_{m-L|m} - \overline{x}_{m-L}) = 0$.*

**8.Section "Simulation and Experiments".** From my point of view, this section

needs to be throughly revised. Where did the nonlinear term came from? Why L=5? What are the values of the initial conditions? And so on.

The major problem in this section is that the figures seem to contract what the authors said in the text. The authors need to explain what exactly is remarkable about the results presented in the paper. The figures showed exactly the contrary and are not convincing (std = 0.0001 - almost no noise contamination - please add a convincing noise threshold).

**Reply:** *Thank you for your helpful suggestion. In the revised version, we have made major changes to the simulation part and corrected the selection of noise. Some simulation results are as follows: case 1 (Gaussian noise): Let the initial condition be zero, that is, $x_0 = 0$ ($\hat{x}_0 = 0$), and suppose the disturbance input $\omega_m$ is $N(0, 1)$. Under the above-mentioned setting conditions, in order to better illustrate the universality that MHE can achieve the goal, we randomly select Gaussian noise and obtain the estimation result using MHE of the system.*

[Figure]

Figure 7: The true measured output $z(m)$ and its estimations $\hat{z}(m)$ based on the MHE with Gaussian noise

*The estimation result of the T-S fuzzy system with Gaussian noise is shown in Figs. 7. Obviously, under the influence of Gaussian noise, the output of the system changes more widely, and the output after adding MHE is more gradual. It shows that when the measured noise satisfies the normal distribution, the performance of estimation is remarkable, and the estimated value curve fluctuates within a smaller range than the true*

value curve.

case 2 (non-Gaussian noise): Let the initial condition $x_0 = 0$ ($\hat{x}_0 = 0$), and assume the disturbance input $\omega_m$ is

$$\omega_m = \frac{3 * sin(0.85m)}{(0.55m)^2 + 1}$$

[Figure]

Figure 8: The noise of T-S fuzzy system in case 2

[Figure]

Figure 9: The true measured output $z(m)$ and its estimations $\hat{z}(m)$ based on the MHE with function noise

The simulation results are shown in Figs. 8 and 9. Fig. 8 is the noise, obviously, the external interference is bounded and non-Gaussian. And Fig. 9 shows the simulation

*run for the T-S fuzzy system with the MHE filter. The proposed MHE can effectively counteract the influence of the sine-form noise in the T-S fuzzy system. In this case, the noise decays with time, and the estimation performance of the MHE is most pronounced during the initial period. A clear improvement of the smoothness can be observed for the T-S fuzzy system, which is the result of the MHE filter reducing noise.*

*case 3 (non-Gaussian noise): To make our proposed MHE estimation scheme more convincing in systems with unknown dynamics, we add a case when the noise is uniformly distributed.*

[Figure]

Figure 10: The noise of T-S fuzzy system in case 3

[Figure]

Figure 11: The true measured output $z(m)$ and its estimations $\hat{z}(m)$ based on the MHE with uniform noise

    *The added noise in this case is shown in Fig. 10 and the effect of the designed MHE filter is shown in Fig. 11. It can be seen from the figure that adding the MHE filter to the T-S fuzzy control model with uniformly distributed noise can make the output smoother. To increase the convincing power, a uniformly distributed noise is added to the designed multi-threaded control system, and MHE filtering is used. It can be seen from the simulation figures that the proposed estimator can work well in systems with unknown factors.*